# Proteomics of *Paracoccidioides lutzii*: Overview of Changes Triggered by Nitrogen Catabolite Repression

**DOI:** 10.3390/jof9111102

**Published:** 2023-11-12

**Authors:** Vanessa Rafaela Milhomem Cruz-Leite, André Luís Elias Moreira, Lana O’Hara Souza Silva, Moises Morais Inácio, Juliana Alves Parente-Rocha, Orville Hernandez Ruiz, Simone Schneider Weber, Célia Maria de Almeida Soares, Clayton Luiz Borges

**Affiliations:** 1Department of Biochemistry and Molecular Biology, Institute of Biological Sciences II, Federal University of Goiás, Goiânia 74690-900, GO, Brazil; andre.bio.br@hotmail.com (A.L.E.M.); lanaohara.loss@gmail.com (L.O.S.S.); moises.biomed@gmail.com (M.M.I.); juparente@ufg.br (J.A.P.-R.); cmasoares@gmail.com (C.M.d.A.S.); 2Estácio de Goiás University Center—FESGO, Goiânia 74063-010, GO, Brazil; 3MICROBA Research Group, Cellular and Molecular Biology Unit, Department of Microbiology, School of Microbiology, University of Antioquia, Medellín 050010, Colombia; orville.hernandez@udea.edu.co; 4Faculty of Pharmaceutical Sciences, Food and Nutrition, Federal University of Mato Grosso do Sul, Campo Grande 79304-902, MS, Brazil; simone.weber@ufms.br

**Keywords:** proteomics, nitrogen starvation, metabolic reprogramming, NCR-proline, *Paracoccidioides*

## Abstract

Members of the *Paracoccidioides* complex are the causative agents of Paracoccidioidomycosis (PCM), a human systemic mycosis endemic in Latin America. Upon initial contact with the host, the pathogen needs to uptake micronutrients. Nitrogen is an essential source for biosynthetic pathways. Adaptation to nutritional stress is a key feature of fungi in host tissues. Fungi utilize nitrogen sources through Nitrogen Catabolite Repression (NCR). NCR ensures the scavenging, uptake and catabolism of alternative nitrogen sources, when preferential ones, such as glutamine or ammonium, are unavailable. The NanoUPLC-MS^E^ proteomic approach was used to investigate the NCR response of *Paracoccidioides lutzii* after growth on proline or glutamine as a nitrogen source. A total of 338 differentially expressed proteins were identified. *P. lutzii* demonstrated that gluconeogenesis, β-oxidation, glyoxylate cycle, adhesin-like proteins, stress response and cell wall remodeling were triggered in NCR-proline conditions. In addition, within macrophages, yeast cells trained under NCR-proline conditions showed an increased ability to survive. In general, this study allows a comprehensive understanding of the NCR response employed by the fungus to overcome nutritional starvation, which in the human host is represented by nutritional immunity. In turn, the pathogen requires rapid adaptation to the changing microenvironment induced by macrophages to achieve successful infection.

## 1. Introduction

The *Paracoccidioides* genus consists of five species: *Paracoccidioides brasiliensis*, *Paracoccidioides americana*, *Paracoccidioides restrepiensis*, *Paracoccidioides venezuelensis* and *Paracoccidioides lutzii*. These species are the causative agents of PCM, a human systemic mycosis endemic in and restricted to Latin America, with higher incidence in Brazil [1,2]. Members of the *Paracoccidioides* genus are able to change their morphological form in a temperature-dependent manner, a process called thermal dimorphism. The fungi grow as mycelia under saprobiotic conditions or when cultured at 25 °C; however, in host tissues or when cultured at 36 °C, the mycelia transit to the yeast parasitic form [3]. PCM infection occurs by inhalation of infective propagules of mycelium or conidial forms that reach the host’s pulmonary alveoli and transit to the yeast form. The yeasts can spread to other organs through lymphatic or hematogenous dissemination, leading to the manifestation of PCM disease in different clinical forms [4,5].

Pathogenic fungi have been found in a variety of habitats, behaving as saprophytic, commensal, or parasitic organisms. Their ability to thrive in diverse niches is closely related to their ability to utilize different nutrient sources present in the extracellular milieu, such as carbon and nitrogen, as well as micronutrients [6,7]. This ability is demonstrated during growth in different substrates and, for pathogenic microorganisms, during host–pathogen interactions [8,9], since the ability to acquire and consume the available nutrients in the host site is a prerequisite for pathogen success [10,11].

Microorganisms are able to utilize a range of nitrogenous compounds including inorganic or organic nitrogen sources. This may also be related to their nutritional status, depending on the nitrogen source available [12,13]. Fungi can metabolize nitrogen sources into glutamate and glutamine, with glutamate being responsible for 85% of the nitrogen availability in fungal cells [14,15]. Thus, nitrogen uptake is an important strategy for fungi, as it ensures the biosynthetic activity inside the cells and coordinates the processes for the adaptation and dissemination of the pathogen inside the host [6,11,16].

NCR is a mechanism that ensures the uptake of non-preferential or alternative nitrogen sources, allowing a cell to synthesize appropriate enzymes that are important in the nitrogen-starved state [17]. Non-preferential sources such as peptides, nitrates, nitrites, purines, proteins, free amino acids and other nitrogen compounds trigger NCR by relieving GATA transcription factors that bind to the DNA through a zinc finger domain and control NCR-sensitive gene expression [18,19]. In the *Saccharomyces cerevisiae* model, the GATA transcription factors Gln3 and Gat1 are essential for NCR alleviation and can control many genes important for fungal survival under nitrogen starvation conditions [18,20,21,22,23,24]. Other well-studied NCR model organisms, such as *Aspergillus nidulans* and *Neurospora crassa*, contain the GATA transcription factors AreA and Nit-2, respectively, that ensure NCR alleviation [25,26,27,28]. On the other hand, when preferential nitrogen sources such as glutamine or ammonium are available in the environment, NCR is repressed [17].

Transcriptomic and proteomic studies during nitrogen starvation demonstrate how available nitrogen affects microorganisms. The transcriptional response of *Candida albicans* in the absence or presence of different nitrogen sources revealed an adaptation, where nitrogen starvation regulatory interactions occurred between the ammonium permease *MEP2* gene, the general amino acid permease *GAP1* gene, and the GATA transcription factors *GLN3* and *GAT1* genes [29].

In *S. cerevisiae*, a transcriptional response to 21 different nitrogen sources provided insights into the yeast´s ability to regulate transcriptional mechanisms [30]. Scherens et al. [31] described a microarray analysis in *S. cerevisiae* in response to glutamine, proline and rapamycin treatments, allowing the identification of 91 NCR-related genes. Furthermore, the GATA transcription factors Gln3 and Gat1 were implicated in regulating the expression of the NCR-sensitive gene *GAP*, which is considered an NCR marker [31]. A proteomic approach in *S. cerevisiae* revealed multiple biological pathways involved in the response to different nitrogen sources, entailing proteins participating in energy regulation, amino acid metabolism, translation, and the stress response [32].

Nitrogen uptake is critical to the life cycle of pathogenic fungi. Studies were conducted on plant pathogens such as *Ustilago maydis* [13], *Magnaporthe oryzae* [33], *Fusarium oxysporum* [34,35], *Fusarium verticillioides* [36] and others. However, few studies have been conducted on human pathogenic fungi. Some human pathogenic fungi, such as *Cryptococcus neoformans* and *C. albicans*, are relatively understudied [29,37,38,39,40].

In general, additional research on NCR is needed in relation to other significant human pathogenic fungi. This could provide a better understanding of the mechanisms required for infection establishment and dissemination in host tissues, as well as elucidate vital biological molecules necessary for nitrogen uptake from the host organism. NCR in *Paracoccidioides* spp. was first investigated by our group [41], demonstrating that NCR alleviation occurs in the presence of the non-preferential nitrogen source proline. To elucidate the global fungal metabolic pathway reprogramming in response to nitrogen availability, essential to infection establishment in host tissues, and explore NCR in *P. lutzii*, a proteomic approach was employed.

## 2. Materials and Methods

### 2.1. P. lutzii Growth Conditions

*Paracoccidioides lutzii* yeast cells (ATCC-MYA-826) were utilized for all experiments. The yeast cells were maintained in solid Fava Netto’s medium for 3 days at 36 °C [42]. The yeast cells were then incubated in Brain Heart Infusion (BHI) medium supplemented with 4% (*w/v*) glucose at 36 °C under rotation for 48 h. The cells were centrifuged at 1900× *g* for 10 min at 4 °C and washed twice in phosphate-buffered saline (PBS). The yeast (2 × 10^6^ cells/mL), was inoculated under NCR conditions, as described by Cruz-Leite et al. [41]. In brief, the yeast cells were grown in Yeast Nitrogen Base (YNB) without amino acids and ammonium sulfate (Sigma-Aldrich^®^, St. Louis, MO, USA), supplemented with either 10 mM glutamine or 10 mM proline nitrogen sources plus 2% glucose (*w/v*) for 48 h. Glutamine, a preferential nitrogen source, triggers repressive NCR conditions, while proline, a non-preferential nitrogen source, triggers alleviation of NCR [14].

### 2.2. Viability Analysis of P. lutzii under NCR Conditions

The viability of *P. lutzii* was determined in NCR conditions and in BHI medium supplemented with 4% (*w/v*) glucose at 36 °C under rotation. *P. lutzii* yeast cells were assessed by optical density at 600 nm at the following time points: 0, 24, 48 and 72 h. Under NCR conditions, *P. lutzii* yeast cells cultured for 48 h were stained with 1 µg/mL (*w/v*) of propidium iodide (Sigma-Aldrich^®^, St. Louis, MO, USA) and visualized using an Axiocam MRc-Scope A1 fluorescence microscope (Carl Zeiss^©^, Oberkochen, BW, DE, Germany).

### 2.3. Ethanol Measurement Assay

Ethanol concentration was indirectly determined using an enzymatic detection kit (UV test for ethanol) according to the manufacturer’s instructions (R-Biopharm^©^, Darmstadt, HE, DE, Germany). In turn, ethanol was oxidized to acetaldehyde by the alcohol dehydrogenase enzyme in the presence of nicotinamide adenine dinucleotide (NAD). Acetaldehyde was quantitatively oxidized to acetic acid in the presence of aldehyde dehydrogenase, releasing NADH, which was determined by measuring the absorbance at 340 nm. *P. lutzii* yeast cells were cultivated under NCR conditions for 48 h, and the supernatant obtained from the lysis of 2 × 10^8^ cells/mL was used in this assay. Briefly, the yeast cells were centrifuged and lysed using glass beads and a bead beater apparatus (BioSpec^©^, Bartlesville, OK, USA) in 5 cycles of 30 s. The cell lysate was centrifuged at 22,500× *g* for 10 min at 4 °C, and the supernatant was used for enzymatic assays. The ethanol concentration was determined by the Student’s *t*-test; *p* ≤ 0.05 was considered significant.

### 2.4. RNA Extraction and Quantitative Real-Time RT-qPCR Analysis

NCR-conditioned-*P. lutzii* yeast cells were used for all RT-qPCR experiments. After 48 h of incubation, the yeast cells were harvested, and total RNA was extracted by using TRIzol (TRI Reagent, Sigma-Aldrich^©^, St. Louis, MO, USA) and mechanical cell rupture, using a bead beater apparatus (BioSpec^©^, Bartlesville, OK, USA). Total RNA was reversibly transcribed using the Revertaid M-Mulv Reverse Transcriptase (Applied Biosystems™, Foster City, CA, USA). The, cDNA was used for RT-qPCR using Power SYBR^®^ Green Master Mix (Applied Biosystems™, Foster City, CA, USA) in a QuantStudio 5 Real-Time PCR System (Applied Biosystems™, Foster City, CA, USA). RT-qPCR analysis was performed in biological triplicates. A relative standard curve was generated by pooling each cDNA sample, which was serially diluted (from 1:5 to 1:125). The relative expression levels of the *NCR* genes formamidase (PAAG_03333), arginase (PAAG_01969), urease (PAAG_00954) and γ-glutamyl transpeptidase (PAAG_06130) were calculated using the standard curve method for relative quantification [43]. The tubulin gene (PAAG_03031) was used as an endogenous control, as described by Cruz-Leite et al. [41]. Student’s *t*-test was applied, and *p* ≤ 0.05 was considered as statistically significant. The specific primers used in all RT-qPCR experiments are listed in Appendix A.

### 2.5. P. lutzii Protein Extraction in NCR Conditions

*P. lutzii* yeast cells cultured under NCR conditions for 48 h were used for protein extraction. After the incubation period, the yeast cells were centrifuged at 1500× *g*, resuspended in 50 mM ammonium bicarbonate, pH 8.5, and disrupted using beads and a bead beater apparatus (BioSpec^©^, Bartlesville, OK, USA). The yeast cell lysate was centrifuged at 10,000× *g* for 10 min at 4 °C, and the supernatant was collected and quantified using the Bradford Bio-Rad Protein Dye Reagent Concentrate assay (Bio-Rad^©^, Hercules, CA, USA), using bovine serum albumin (BSA) as a standard [44]. SDS-polyacrylamide gel electrophoresis (SDS-PAGE) was performed on 12% polyacrylamide gels to assess protein integrity.

### 2.6. P. lutzii nanoUPLC-MS^E^ Analysis under NCR Conditions

Proteins acquired under NCR conditions (150 µg) were prepared for nano-UPLC-MS^E^ in line with the procedures outlined by Murad et al. [45] and Parente-Rocha et al. [46], with minor modifications. To each sample, 30 μL of 50 mM ammonium bicarbonate (NH_4_HCO_3_) was added, followed by an additional 75 μL of RapiGEST™ (Waters™, Milford, MA, USA) (0.2% *v*/*v*), and then the samples were incubated for 15 min at 80 °C. Afterward, 100 mM dithiothreitol (DTT) was included, and the samples were allowed to sit for 30 min at 60 °C. After the initial 30 min period, 7.5 μL of iodoacetamide at a 300 mM concentration was added, and the samples were kept at room temperature, shielding them from light. The enzymatic digestion was carried out by adding 30 μL of trypsin to each aliquot, which was incubated at 37 °C for 16 h. To purify the resulting peptides, RapiGEST™ was precipitated by adding 10 μL of 5% TFA and then incubating the solution at 37 °C for 90 min. The samples were later centrifuged at 18,000× *g* for 30 min, after which the supernatants containing the digested peptides were collected. The peptides were transferred to Waters Total Recovery vials (Waters™, Milford, MA, USA). To the vials, 15 μL of rabbit phosphorylase B (Waters™, Milford, MA, USA) was added, along with 255 μL of a solution containing 3% acetonitrile and 0.1% formic acid.

The peptides were then subjected to nanoUPLC-MS^E^ and separated by two-dimensional liquid chromatography using the nanoACQUITY™ M-class system (Waters™, Milford, MA, USA). The initial step involved using a 5 μm UPLC M-Class Peptide BEH C18 column, 130 Å (300 μm × 50 mm—Waters™, Milford, MA, USA) for the first dimension. Subsequently, every fraction eluted was trapped in a 5 μm Acquity UPLC M-Class Symmetry C18 trap column, 100 Å (180 μm × 20 mm—Waters™, Milford, MA, USA). The peptides were separated in the second dimension through the use of an Acquity UPLC M-Class HSS T3 1.8 μm (75 μm × 150 mm) analytical column and were analyzed in triplicate. A solution containing 200 fmol/μL of the precursor ion [Glu1]-Fibronopeptide B human (*m*/*z* 785.8426) (Sigma-Aldrich^©^, St. Louis, MO, USA), was used for mass calibration. The solution was measured every 30 s at a constant flow rate of 0.5 μL/min. The eluted peptides underwent analysis using a Synapt G1 HDMS™ mass spectrometer (Waters™, Wilmslow, CHS, UK). The instrument, called nanoESI-Q-TOF (Waters™, Wilmslow, CHS, UK), included a nano-electrospray ion source and a quadrupole and a time-of-flight analyzers.

The obtained MS^E^ spectra data were processed using the ProteinLynx Global server v.3.0.2 (PLGS) and searched against a *P. lutzii* database (https://www.ncbi.nlm.nih.gov/genome/?term=paracoccidioides, accessed on 27 September 2022) together with generated reverse sequences. The mass error tolerance for peptides identification was set under 50 ppm. A protein detected in all replicates (PAAG_05093), presenting a variance coefficient of less than 10%, was used to normalize the expression data and then compare the protein levels between NCR conditions in the presence of 10 mM glutamine or 10 mM proline nitrogen sources. The protein and peptide tables generated by PLGS were merged, and quality parameters such as dynamic range, peptide type, and mass accuracy were determined for each condition using the software MassPivot v3.1 and FBAT [45].

Proteins presenting expression value differences of 30% between the NCR samples were considered regulated. The Spotfire^®^ v.7.0.0 software was used to compare the peptide and protein tables and to create graphics for all data. The Uniprot (http://www.uniprot.org/uniprotkb?query=paracoccidioides, accessed on 27 September 2022) database was used for functional classification, while the NCBI database (https://www.ncbi.nlm.nih.gov/protein/?term=Paracoccidioides, accessed on 27 September 2022) was used for the annotation of uncharacterized proteins. The proteins up-regulated under NCR conditions underwent adhesin prediction by using the FaarPred database (https://bioinfo.icgeb.res.in/faap/team.html, accessed on 27 September 2022), with default software parameters (e-value ≥ −0.8) [47].

### 2.7. Enzymatic Activities under NCR Conditions

The activities of formamidase, γ-glutamyl transpeptidase, and glutathione S-transferase were assessed in extracts from *P. lutzii* yeast cells incubated under NCR conditions. After 48 h, the yeast cells were recovered from the NCR conditions, and total protein extracts were obtained and submitted to the analysis of enzymatic activities. For formamidase activity, 10 μg of the *P. lutzii* total protein extract was used, as previously described by Borges et al. [48]. In brief, the amount of ammonia released from each sample was determined by comparison with a standard curve. One unit (U) of formamidase specific activity corresponds to the amount of enzyme required to hydrolyze 1 µmol of formamide (corresponding to the formation of 1 µmol of ammonia) per min per mg of total protein. We performed the γ-glutamyl transpeptidase assay (GGT) as described by Silber et al. [49], with modifications. In brief, amounts of 4 and 12 μg of proteins from NCR-proline and NCR-glutamine conditions, respectively, were used in the assay. GGT activity measures the rate at which γ-glutamyl-ρ-nitroanilide (GPNA) is cleaved to generate ρ-nitroaniline (ρNA), whose absorbance is measured at 405 nm. One unit (U) of GGT releases 1 µmol of ρNA per minute. The number of nmol of pNA released in the samples was calculated using a standard curve. GGT specific activity was assessed by determining the number of nmol of ρNA released per liter of sample in the incubation time (min). To determine glutathione S-transferase activity, a glutathione S-transferase assay kit (Sigma-Aldrich^®^, St. Louis, MO, USA) was used, analyzing 10 μg of protein under both NCR-proline and NCR-glutamine conditions. This assay involves the conjugation of L-glutathione to CDNB (1-chloro-2,4-dinitrobenzene) through the thiol group of glutathione via the action of glutathione S-transferase. The product generated, the GS–DNB conjugate, absorbs at 340 nm, and the increase in the absorption rate is directly proportional to the enzyme activity in the sample.

### 2.8. Estimation of Cell Wall Components in P. lutzii

Recovered *P. lutzii* yeast cells under NCR conditions were utilized for the analysis of the cell wall components chitin and glucans. The yeast cells were fixed in 100% methanol at −80 °C for 20 min and then at −20 °C for another 20 min, subsequently washed twice in PBS 1X with centrifugation and then used for the analysis. The calcofluor white (CFW) method (Sigma-Aldrich^®^, St. Louis, MO, USA) was employed for chitin analysis. The cells were stained with CFW (100 µg/mL in PBS 1X) for 30 min and washed with PBS 1X, as indicated by the manufacturer. The analysis of beta glucan staining was conducted using a 100% aniline blue solution (Sigma-Aldrich^®^, St. Louis, MO, USA). The yeast cells were incubated with aniline blue for 5 min under stirring and subsequently washed twice in PBS 1X. The samples stained with either CFW or aniline blue were visualized using an Axiocam MRc-Scope A1 fluorescence microscope (Carl Zeiss^©^, Oberkochen, BW, DE, Germany), and fluorescence intensity was measured using the AxioVision Software, version 4.8.2.0 (Carl Zeiss^©^, Oberkochen, BW, DE, Germany). A minimum of 100 cells for each microscope slide were examined to evaluate fluorescence intensity, in triplicate for each NCR sample. The AxioVision software determined the fluorescence intensity (in pixels) and standard error in each analysis. Statistical comparisons were conducted using the by Student’s *t*-test, with statistical significance set at *p* ≤ 0.05.

### 2.9. Ex Vivo Model of Infection in NCR Conditions

*P. lutzii* yeast cells cultivated in liquid BHI medium and pretreated in NCR conditions were used for the evaluation of their internalization and survival inside macrophages. J774 macrophages (Rio de Janeiro Cell Bank-BCRJ/UFRJ, accession number 0273) were maintained in Dulbecco’s Modified Eagle Medium (DMEM) (Vitrocell-Embriolife^©^, Campinas, SP, BR) supplemented with 10% (*v*/*v*) bovine fetal serum (Vitrocell-Embriolife^©^, Campinas, SP, BR). A total of 10^6^ macrophages/mL were plated onto a 12-well polypropylene plate containing DMEM and 1 U/mL of IFN-γ (Sigma-Aldrich^©^, St. Louis, MO, USA) and incubated for 12 h at 36 °C and in 5% CO_2_ to activate the macrophages. Prior to the ex vivo assays, *P. lutzii* yeast cells were grown in liquid BHI medium for 48 h at 36 °C, pre-treated in NCR conditions (10 mM proline and 10 mM glutamine) for 48 h at 36 °C and then recovered, washed twice in PBS 1X and incubated in fresh DMEM containing IFN-γ (1 U/mL). A total of 5 × 10^6^ *P. lutzii* yeast cells/mL was added to 1 × 10^6^ cells/mL macrophages, at a multiplicity of infection (MOI) of 1:5 (macrophages/*P. lutzii* yeast cells). The cells were incubated for 24 h at 36 °C with 5% CO_2_. Afterwards, the macrophages were lysed with cold water, and the fungal cells were recovered as described previously by Lima et al. [50]. The number of viable cells was determined by quantifying the number of colony-forming units (CFU), and the data are expressed as the mean value ± standard deviation of triplicates. Statistical analyses were performed using the Student’s *t*-test.

## 3. Results

### 3.1. P. lutzii Behavior under NCR Conditions

Yeast cells of *P. lutzii* were grown under NCR conditions in YNB with either 10 mM glutamine or 10 mM proline to assess the impact of the NCR treatments. *P. lutzii* under NCR conditions demonstrated a slow growth, compared to the its growth in nutrient-rich BHI medium (Appendix A). However, *P. lutzii* yeast cells exhibited comparable fitness when grown in proline or glutamine (Appendix A). The yeast cells recovered from the NCR conditions within 48 h were stained with propidium iodide (1 µg/mL), demonstrating that the cells were viable (Appendix A). These findings suggested that *P. lutzii* cells have the capability to adapt and survive in a nitrogen starvation environment.

### 3.2. Proteomic Data Quality Analysis

According to Cruz-Leite et al. [41], the NCR response is achieved in 48 h, as evidenced by an increase in *gap1* transcripts, an NCR marker. Therefore, we chose this incubation time for further comparative proteomic analysis. NanoUPLC-MS^E^ was conducted, following a previously outlined methodology [51], and the protein and peptide data generated were processed using PLGS, as depicted in Appendix A. Under NCR-proline conditions, 42.3% of the peptides were obtained from peptide match type data in the first pass, and 10.4% in the second pass. Additionally, 11.3% of the peptides were identified by a missed trypsin cleavage, and there was a 24.7% rate of in-source fragmentation, as shown in Appendix A. On the other hand, under NCR-glutamine conditions, 51.2% of the peptides were obtained from peptide match type data in the first pass, and 9% in the second pass. A total of 10.7% of the peptides were identified by a missed trypsin cleavage, and a rate of 19.2% in source fragmentation was observed, as shown in Appendix A. The dynamic range detection is shown in Appendix A and demonstrated a good detection distribution over three orders of magnitude for protein concentrations in NCR-proline and NCR-glutamine conditions. In Appendix A, peptide mass accuracy analyses were assessed, whose peptide data were used to generate a bar graph showing the mass accuracy for peptides in each condition. The majority of the peptides in both NCR-proline and NCR-glutamine conditions had an error of less than 10 ppm, indicating high-quality data.

### 3.3. Proteomic Analysis

The analysis of the *P. lutzii* proteome was conducted by comparing yeast cells that were grown under NCR-proline to cells grown under NCR-glutamine conditions. Cells cultured in the presence of the preferential nitrogen source glutamine were selected as the control. A total of 439 proteins were identified. We employed a 1.3-fold change as a cutoff to determine the regulated proteins. Proteins presenting fold changes less than or equal to 0.69 were considered down-regulated, while proteins with fold changes ranging from 0.7 to 1.29 were considered constitutively expressed, and proteins presenting fold changes up to 1.3 were considered up-regulated. In total, 338 proteins were identified as differentially expressed, as reported in Appendix A, revealing 269 up- and 69 down-regulated proteins, respectively. All proteins were categorized according to the Uniprot database, as illustrated in Figure 1, which displays biological processes that were either enhanced or inhibited under the analyzed NCR conditions. The proteins grouped into the metabolism category were the most represented after 48 h under NCR conditions (Appendix A). Up-regulated proteins related to metabolism were involved in various metabolic pathways, including amino acid metabolism, nucleotide/nucleoside/nucleobase metabolism, C-compound/carbohydrate and fatty acid biosynthesis and lipid metabolism. Other categories exhibited significant up-regulated proteins, such as those involved in cell rescue, defense and virulence and energy regulation, emphasizing the following processes: detoxification, stress response, glycolysis and gluconeogenesis, pentose phosphate pathway, glyoxylate cycle, methylcitrate cycle and fatty acid oxidation.

### 3.4. An Overview of P. lutzii Metabolism under NCR Conditions

The *P. lutzii* response to NCR conditions is illustrated in Figure 2, outlining the fungus’ metabolism, energy regulation, and adaptation to the NCR-proline conditions. In summary, the glycolysis pathway and tricarboxylic acid cycle were downregulated, and the pyruvate shunt produced by amino acid degradation appeared to promote gluconeogenesis, resulting in oxaloacetate production through the induction of pyruvate carboxylase. In this sense, the gluconeogenesis shunt could be driven by oxaloacetate, a key intermediate of this pathway and up-regulated in NCR-proline condition. The first regulatory step of gluconeogenesis requires acetyl-CoA, produced by β-oxidation, as a positive effector (Figure 2). The proteomic data suggested that, in NCR conditions, *P. lutzii* increased gluconeogenesis, while decreasing glycolysis.

Acetyl-CoA production is fed by β-oxidation, which involves up-regulated enzymes including carnitine O-acetyltransferase, acyl-CoA dehydrogenase and 3-ketoacyl-CoA thiolase. However, it appeared that acetyl-CoA was not utilized in the tricarboxylic acid cycle, which was identified as down-regulated (isocitrate dehydrogenase and succinate dehydrogenase). Acetyl-CoA can shunt to the glyoxylate cycle, generating glyoxylate and succinate molecules through the up-regulated enzymes isocitrate lyase and malate synthase (Appendix A). In this sense, the glyoxylate cycle could have the potential to be utilized in NCR-proline conditions to produce glucose for *P. lutzii* yeast cells. Propionyl-CoA, the substrate for odd-chain lipid β-oxidation, can be shunted to the methylcitrate cycle via the up-regulation of the enzyme exclusive to this pathway, 2-methylcitrate dehydratase (Appendix A). In summary, acetyl-CoA production would be important for energy production in NCR-proline conditions, providing intermediary molecules such as NADH_2_ and FADH_2_ for the electron transport chain for subsequent ATP synthesis.

The anaerobic pathway for producing ethanol was also activated under NCR-proline conditions via alcohol dehydrogenase enzyme induction, as shown in Figure 2 and Appendix A. Consistent with the proteomic data, an enzymatic ethanol measurement was conducted, and the findings revealed noteworthy variations in the ethanol pathway after 48 h under NCR-proline conditions (Figure 3).

In NCR-proline conditions, the enzymes glucose-6-phosphate1-dehydrogenase and 6-fosfogluconate dehydrogenase were regulated, inducing the oxidative phase of the pentose phosphate pathway that provides NADPH-reducing intermediates for biosynthetic pathways. Ribose-phosphate pyrophosphokinase triggered the production of D-ribulose-5-phosphate, which acts as a precursor substrate for purine and pyrimidine metabolism. Amino acid metabolism was observed to be prominent in NCR-proline conditions, where the pathways associated with amino acid biosynthesis and degradation appeared to be intensely activated, allowing adaptation to nitrogen starvation (Appendix A). The arginine biosynthetic pathway contains NCR-regulated enzymes such as glutamine synthetase and NADP-specific glutamate dehydrogenase, while the glutamate degradation pathway involves the NCR-regulated enzyme delta-1-pyrroline-5-carboxylate dehydrogenase.

### 3.5. Proteins Potentially Involved in Adhesion during the NCR Response

To examine adherence-related proteins under NCR conditions, the pathogenic fungi prediction tool FaaPred was employed. Appendix A displays 56 up-regulated proteins in NCR-proline conditions that were predicted to be related to adherence processes. The proteins predicted to be associated with adhesion were also involved in other biological processes, such as glycolysis/gluconeogenesis (alcohol dehydrogenase), β-oxidation (3-ketoacyl-CoA thiolase B), amino acid metabolism (aminomethyltransferase), translation (actin cytoskeleton protein VIP1), defense and virulence (urease accessory protein, carbonic anhydrase), detoxification (thioredoxin) and stress response (peroxidase). These results could suggest that *P. lutzii* may use these molecules to increase its attachment during host–pathogens interactions.

### 3.6. Regulation of Cell Wall Metabolism in P. lutzii

Proteins responsible for cell wall synthesis or maintenance were also investigated under NCR conditions. In NCR-proline conditions, cell wall remodeling was observed, as evidenced by the analysis of the cell wall components glucan and furanose (Figure 4). The amount of 1,3-beta-D-glucan polymer in the cell wall was estimated using aniline blue. Fluorescence intensity visibly increased for the cells cultured under NCR-proline conditions (Figure 4a), consistent with the quantitative analysis of fluorescence intensity (Figure 4b). The up-regulation of the mannosyl-oligosaccharide glucosidase enzyme (Appendix A) can be associated with the observed increase in glucan content under NCR-proline conditions. The evaluation of changes in the cell wall was conducted using the CFW fluorophore. Figure 4c,d show a significant increase in CFW fluorescence intensity (*p* ≤ 0.05) under NCR-proline conditions. The presence of the UDP-galactopyranose mutase (Appendix A) indicated the production of furanose in yeast cells cultured in proline conditions.

### 3.7. Transcriptional Level Analysis of NCR-Related Genes

NCR alleviation leads to an increase in the expression of NCR-related genes responsible for scavenging, uptake, and catabolism of non-preferential nitrogen sources [17]. Figure 5 shows that the transcriptional level of the genes encoding formamidase, arginase, and γ-glutamyl transpeptidase increased by approximately 30% under NCR-proline conditions compared to NCR-glutamine conditions. The transcriptional levels of urease increased by approximately 12% under NCR-proline conditions compared to NCR-glutamine conditions. Thus, it can be concluded that proline as a nitrogen source had an impact on the transcriptional levels of NCR-related genes in *P. lutzii* yeast cells.

### 3.8. Protein Level Confirmation Analysis under NCR Conditions

To examine processes related to cell defense responses in NCR conditions, which were demonstrated by the up-regulation of enzymes pertaining to the biological category of cell rescue, defense and virulence, the enzymatic activities of gamma-glutamyltransferase and glutathione S-transferase proteins were evaluated (Figure 6 and Appendix A). The enzyme formamidase, which exhibited a 2.48-fold increase in our proteomic data, was also regulated by NCR mechanisms (Appendix A). In NCR-proline conditions, the enzymatic activity of formamidase was significantly greater than in NCR-glutamine conditions (Figure 7).

### 3.9. Influence of NCR Training in P. lutzii Survival Inside Macrophages

The survival capacity of *P. lutzii* yeast cells inside macrophages was assessed following training cultivation in NCR and rich culture medium (BHI) conditions. *P. lutzii* yeast cells were cultured in proline and glutamine conditions for 48 h, before being challenged with activated J774 macrophages. This study aimed to replicate a host starvation environment, specifically, macrophage conditions following pathogen contact and internalization [52]. Figure 8 depicts that, after 24 h of cell internalization in proline culture, *P. lutzii* yeast cells exhibited greater resistance to macrophage-mediated killing when compared to cells cultured in BHI or glutamine conditions.

## 4. Discussion

Nitrogen catabolite repression (NCR) was observed in members of the *Paracoccidioides* complex, indicating the presence of the GATA transcription factor AreA, which ensures the synthesis of enzymes necessary for the scavenging, uptake and catabolism of non-preferential or alternative nitrogen sources [41]. The activation of NCR is triggered by GATA transcription factors that detain a Zinc finger motif for complete functionality [17]. The induction of NCR is influenced by the presence of non-preferential nitrogen sources, particularly proline, which is the most widely used amino acid for NCR response assays [14,30,53]. This study presented a comprehensive overview of the NCR-proteome in *P. lutzii*, revealing the protein network involved in NCR inside yeast cells.

In this proteomic study, the amino acid proline was utilized as the NCR response trigger in *P. lutzii*, while the amino acid glutamine was used as a control in all experiments. It is worth noting that glutamine is considered a preferred nitrogen source that represses NCR. In *S. cerevisiae*, this repression affects the dephosphorylation pathway and the entry of the GATA transcription factor activator Gln3 into the nucleus [54,55]. Glutamine is the most abundant free amino acid in human blood, reaching concentrations of 2.5 mM [56]. The growth and adaptation mechanisms of *P. lutzii* yeast cells in NCR-proline conditions reflect the modulation of fungal metabolism to ensure cell survival in nitrogen starvation conditions. Furthermore, the yeast cells acquired the ability to retain viability for extended periods under starvation conditions, showing a stable behavior.

In vivo, nutritional deprivation conditions are induced by the defense mechanisms of human cells upon contact with fungal cells. The fungus modulates the activity of the immune system, which is composed of phagocytic cells, such as macrophages and neutrophils, that are then recruited and serve as the first line of defense against microbial infections [57]. Thus, the phagosome formed by phagocytic cells constitutes a hostile environment characterized by acid pH, hydrolytic enzymes and nutrient/nitrogen starvation or limitation [52]. *C. albicans* responds metabolically upon internalization by macrophages, which is important for the fungus survival at its first encounter with the immune system. This entails remodeling its metabolism from glycolysis to gluconeogenesis, generating acetyl-CoA through the β-oxidation of fatty acids and activating the glyoxylate cycle [57]. In addition, the glyoxylate cycle ensures the utilization of compounds containing two carbons to satisfy the cellular carbon requirements and provides a route for the biosynthesis of macromolecules from C_2_ compounds, which can serve as a carbon source in gluconeogenesis [58]. In a mouse model of systemic candidiasis, a mutant strain of *C. albicans* lacking the regulatory enzyme of the glyoxylate cycle isocitrate lyase (Δ*icl1*/Δ*icl1*) lost the ability to use C_2_ carbon sources. Thus, remarkably during phagocytosis by macrophages, the glyoxylate cycle was found to be necessary for virulence [59]. Barelle et al. [60] showed that the glyoxylate cycle is also important for *C. albicans* for protection against host antimicrobial defenses and to facilitate the anabolic metabolism in the absence of fermentable carbon sources. The induction of the glyoxylate cycle in *P. brasiliensis* after phagocytosis by host cells highlighted the yeast mechanisms for adaptation to the harsh milieu of macrophages, which is a hostile environment [61].

Studies of *Paracoccidioides* spp. in the presence of carbon starvation, propionate or sodium acetate and in other environments that mimicked starvation highlighted the metabolic reprogramming that occurred depending on carbon availability and sources, demonstrating the strategies adopted by the fungus for its survival, which included the production of enzymes required for fermentation, such as alcohol dehydrogenase, and β-oxidation, such as 3-ketoacyl-CoA thiolase [50,62]. However, β-oxidation can produce acetyl-CoA and propionyl-CoA, the latter being a toxic compound that results from the degradation of odd-chain fatty acids and branched-chain amino acids. The efficient degradation of propionyl-CoA is driven by the methylcitrate pathway, which produces succinate and pyruvate molecules, which can be mobilized for energy and biomass production [63,64]. The methylcitrate cycle was induced and necessary for virulence traits in *P. brasiliensis* under propionate stimulus [65]. In host tissues, *P. brasiliensis* uses lipids as an energy source during the initial stages of pulmonary infection and exhibits significant ethanol production, as demonstrated by in vivo experiments, in line with the ex vivo response of *Paracoccidioides*. To reinforce this finding, the impact of ethanol production on virulence was extensively investigated in ex vivo models using pathogenic microorganisms [46,66].

Given the difficulty of surviving inside macrophages, an analysis of *P. brasiliensis* proteome demonstrated the involvement of glucose production via gluconeogenesis, potentially fueled by amino acid catabolism, and the enhancement of β-oxidation, as shown by the presence of regulatory enzymes typical of this pathway, in addition to the production of ethanol [46]. Lima et al. [50] also reported such a response when *P. brasiliensis* yeast cells were subjected to carbon starvation. However, infection assays performed using macrophages and *P. brasiliensis* cells pre-exposed to carbon starvation, indicated that the fungus lacked the ability to survive inside the macrophages [50]. Otherwise, the *P. lutzii* yeast cells, when pre-exposed to a non-preferential nitrogen source in NCR-proline conditions, demonstrated an enhanced ability to survive inside macrophage cells. This indicated that immune cells’ capacity to kill the pathogen was impaired. Remarkably, the protein set released under NCR-proline conditions was able to enhance the protection of the fungus against this hostile environment.

*P. lutzii* appeared to utilize enzymes involved in gluconeogenesis, β-oxidation, glyoxylate cycle, methylcitrate cycle and fermentation to thrive in NCR-proline conditions, which mimicked the hostile macrophages/phagosome environment. *P. lutzii* cells’ energy requirements under NCR-proline conditions could be met by alcoholic fermentation and an increase in β-oxidation. In fact, the analysis of the NCR-proline proteome of *P. lutzii* indicated that the proteins identified were involved in multiple functions, including metabolism, energy regulation, oxidative stress, cell rescue, defense and virulence, among others.

Proteins involved in maintaining the intracellular redox state and protecting against oxidative stress, including superoxide dismutase, peroxiredoxin, catalase, thioredoxin, and glutathione S-transferase, were also identified in the *P. lutzii* yeast cells proteome under NCR-proline conditions. In *Paracoccidioides* spp., various stress responses mimicking the host environment were demonstrated, such as hypoxia [67], micronutrients deprivation [68,69], carbon starvation [50], oxidative stress [70], as well as osmotic stress [71]. An efficient response to oxidative stress is essential for the survival of pathogens inside phagosomes. This response is mediated by the presence of reactive oxygen species (ROS)-scavenging enzymes, such as superoxide dismutase, thioredoxins, cytochrome C peroxidase and catalases, that protect the *P. brasiliensis* fungus against the stress generated by the host’s defense mechanisms [70]. In *Paracoccidioides*, reactive oxygen species can also be generated by endogenous events, such as the functional mitochondrial electron transport chain and β-oxidation [72]. In *Histoplasma capsulatum*, the activity of the enzyme superoxide dismutase is linked to the pathogenesis and virulence inside phagocytes during histoplasmosis infection [73]. This study on virulence strategies reinforces the finding that *P. lutzii* produces molecules to ensure its redox maintenance in NCR-proline conditions that mimic the environment in the host tissues.

Another antioxidant defense employed by fungi is the utilization of glutathione, a protective agent against oxidative stress damage, which reacts non-enzymatically with many reactive oxygen species, maintaining the cytosol of the cell in a more reduced state [74]. An additional property of glutathione is its function as alternative nitrogen source in nitrogen starvation conditions, triggered by the enzyme gamma-glutamyl transpeptidase (γ-GT or GGT), which presents a high transcriptional level in *S. cerevisiae* cultured under proline as a nitrogen source [75]. The presence of gamma-glutamyl transpeptidase at the transcriptional and proteomic levels, as well as its enzymatic activity in *P. lutzii* yeast cells grown in NCR-proline sources, reflects the cellular requirements for nitrogen uptake and utilization under nitrogen-restricted conditions.

In *P. lutzii*, the stress response under NCR-proline conditions is represented by the abundance of many heat shock proteins (HSPs). HSPs are regulated in many stress conditions, playing crucial roles in cellular adaptation to various changes, including temperature, osmotic pressure, pH changes and oxidative stress [76,77]. In *Paracoccidioides*, HSP 90 is pivotal for the initial interaction with macrophages, thereby enhancing the fungus’s defenses against exogenous oxidative and acidic stress [78].

*Paracoccidioides* spp. present four genes that encode carbonic anhydrase, a metalloenzyme involved in the reversible hydration of CO_2_ that generates a proton and ammonium bicarbonate HCO_3_^−^, which demonstrate regulatory patterns according to the morphological phase of the fungus [79]. The carbonic anhydrase transcripts *ca2* and *ca4* are highly induced in yeast phase [79], and *ca1* is induced in iron-limited conditions [69]. In an in vivo mouse model of PCM, the transcriptional level of carbonic anhydrase was highly induced in the liver and spleen at 7 and 15 days post-infection, demonstrating its importance for the fungus survival inside these organs [79]. In this sense, the authors suggested that the presence of carbonic anhydrase provided bicarbonate for malonyl-CoA synthesis, which is an important precursor for lipid synthesis. The carbonic anhydrase Ca2 was only detected in NCR-proline conditions, and the carbonic anhydrase Ca1 was up-regulated in *P. lutzii* yeast cells, wherein carbonic anhydrase could be involved in fungi adaptation to the microenvironment.

The formamidase gene of *A. nidulans* is regulated by NCR and presents a GATA motif in the promoter region [80]. The role of formamidase in the virulence of *P. brasiliensis* is under intensive investigation due to its high prevalence in the dimorphic phases of fungi, as shown by in vitro and ex vivo omics experiments [46,51]. Due to its association with determinants of virulence and nitrogen metabolism in *Paracoccidioides* spp., Silva et al. [81] employed antisense RNA technology to silence the formamidase gene in *P. lutzii*. The knockdown of formamidase in *P. lutzii* cells showed a negative effect on the fungus’s survival within macrophages. The expression of formamidase under NCR-proline conditions could be important to increase nitrogen availability or as a virulence determinant, which is currently under investigation by our research group.

The enzymes involved in the arginine biosynthetic pathway were shown to feed the urea shunt in NCR-proline conditions, which serves as a potential source of nitrogen in nitrogen-starved environments. Arginase, a catabolic enzyme, is NCR-regulated by the transcription factor Gln3 in *S. cerevisiae* [82] and is also regulated in *P. lutzii* in NCR-proline conditions. Urea serves as a substrate for urease, an enzyme described as a virulence factor in many pathogens [83,84,85,86,87]. It was previously reported that the enzyme plays a crucial role in nitrogen assimilation when the host–pathogen interaction leads to micronutrient limitation [88,89,90]. In addition, the knockout of the *AreA* gene induces fungal growth defects upon exposure to various nitrogen sources, with the exceptions of glutamine and ammonia [91]. Urease is critical for the growth of *A. fumigatus*, urea being the sole nitrogen source of this fungus. In studies carried out by Xiong et al. [92], there was an increase in the transcriptional level of urease mRNA after *A. fumigatus* was grown in urea media. It was demonstrated that a urease mutation in *S. cerevisiae* led to an increased expression of NCR-regulated genes under conditions of repression [93,94,95].

The response of *P. lutzii* to NCR-proline revealed a correlation with alterations in cell wall components. During the early stages of infection, the cell wall plays a crucial role in host–pathogen interactions and contributes to fungal adaptation to different environmental stresses [96]. Chitin and β-glucans are the most common polysaccharides in fungal species and are recognized by the host immune system. In *A. fumigatus*, the presence of chitin and β-glucans in the cell wall is able to diminish the immune response in both in vivo and in vitro models [97]. Glucans, which are structural polymers of the cell wall and possess immunomodulatory activities, can activate leukocytes and phagocytes and contribute to cytotoxic and antimicrobial activities [98]. According to Felipe et al. [99], *Paracoccidioides* spp. preferentially encode β-glucans during the mycelium phase, whereas in the yeast phase they encode α-glucans. Interestingly, yeast cells that promote the reduction of immunogenic polysaccharide β-glucans and their replacement with α-glucans in the cell wall use strategies for survival inside the host, in which the recognition of the yeast cells by phagocytic cells in the host is hampered by masking mechanisms [100]. This was observed in *H. capsulatum*, with changes in the glucan content, which augmented the opportunistic fungus’s virulence attributes [101]. Additionally, *P. lutzii* demonstrated the production of furanose in NCR-proline conditions. In *A. fumigatus*, galactofuranose serves as a significant constituent of the cell wall and surface glycans. The deletion of the UGM gene, which encodes UDP-galactopyranose mutase, reduces virulence, promotes cell morphology defects and increases the fungus’s susceptibility to antifungal agents [102,103,104].

The understanding of adhesion mechanisms and nitrogen starvation-induced pathways is limited. Thus, this study investigated the correlation between NCR conditions and the mechanisms utilized by the fungus for dissemination. Our results indicated that 20.7% of the up-regulated proteins in NCR-conditions were predicted to be adhesins and that they could be involved in different biological processes, including metabolic pathways, protein synthesis, detoxification, the stress response, defense mechanisms and virulence. Among the predicted adhesins, we identified alcohol dehydrogenase (PAAG_00403), which binds to plasminogen in *C. albicans* [105]. Additionally, this enzyme was identified in extracellular vesicles of *S. cerevisiae* [106] and *H. capsulatum* [107]. In *Paracoccidioides* spp., this fermentation enzyme was identified in the cell wall proteome of *P. lutzii*, along with another alcohol dehydrogenase (PAAG_04541) [108], which is also secreted by *P. restrepiensis* [109].

Additionally, thioredoxin, an enzyme involved in detoxification and predicted to be an adhesin, binds to plasminogen in *C. albicans* [105]. An important aspect for the interaction with the host is that this cytosolic enzyme was also found in extracellular vesicles of *P. brasiliensis* [110] and *C. neoformans* [111], was secreted by *P. brasiliensis* [109], was produced by *P. lutzii* interacting with macrophages [112] and was identified in the cell wall proteome of *Paracoccidioides* spp. [108,112,113].

The actin cytoskeleton protein VIP1 was predicted to be an adhesin in this study, as well as in a previous analysis of the cell wall proteome of *Paracoccidioides* spp. [113]. This protein was also identified in the cell wall proteome of *P. lutzii* [108] and in extracellular vesicles of *P. brasiliensis* [110], suggesting an atypical cellular localization for this nuclear protein. This protein was identified among the proteins of *P. lutzii* that interact with macrophages [112] and was induced after *P. lutzii* interaction with activated macrophages [114]. Regarding adhesins, the discovery of aminomethyltransferase’s involvement in amino acid metabolism is noteworthy. In *P. lutzii*, this mitochondrial enzyme is secreted [115] and binds to plasminogen [116], suggesting that it might be important during host–pathogen interactions. Intracellular proteins present in the cell wall, secreted or localized inside extracellular vesicles might be moonlighting proteins. Moonlighting proteins are able to exert different functions depending on their subcellular localization. Moonlighting proteins typically possess adhesive properties and, when secreted, bind with host components, contributing to adhesion, internalization and invasion of the pathogen [117]. The *P. lutzii* proteins mentioned earlier are related to intracellular pathways; however, according to bioinformatics predictions and the literature, they can also be found in the cell wall and/or extracellularly and/or can even bind to plasminogen. This suggests that they may have an additional functional role in NCR-proline conditions related to fungal attachment, which makes them potential moonlighting proteins.

Amino acids metabolism was identified as more induced under NCR-proline conditions. Cruz-Leite et al. [41] reported that numerous genes are involved in amino acid biosynthetic and degradation pathways, which are essential components of fungal virulence due to their role in nutrient acquisition from the host and in metabolic flexibility. Yeast cells induce a catabolic process to supply essential amino acids for the production of nitrogen molecules during adaptation to nitrogen starvation in their environment [118]. As a result, the nitrogen catabolic process promotes the renovation of necessary proteins for cellular adaptation. This process is sustained by the nitrogen donors glutamate and glutamine, which are produced by the efficient degradation of all-nitrogen compounds and any carbon source inside the cell, where nitrogen accessibility regulate metabolism, transcription, translation, growth and protein turnover [6,16].

## 5. Conclusions

The proteomic approach used for *P. lutzii* under proline and glutamine conditions was applied for the first time in this study to analyze NCR in our organism model. The response to NCR conditions showed fungal metabolic reprogramming in nitrogen stress conditions. In this study, we mimicked the stress generated by host tissues. We displayed the regulation of numerous metabolic pathways that are necessary for fungus vitality, viability, dissemination and survival. The *P. lutzii* fungus was shown to implement several adaptations, including replacing glycolysis with gluconeogenesis, enhancing β-oxidation, regulating the glyoxylate cycle and the methylcitrate cycle and remodeling its cell wall, to compensate for nitrogen starvation. Furthermore, we demonstrated the fungus’s ability to survive within macrophages after pre-exposure to NCR-proline conditions. This study identified potential molecules and pathways involved in host–pathogen interactions that may contribute to the survival and dissemination of fungi within host tissues. The NCR system is believed to trigger these interactions, but further investigation is required to fully understand the relationships between these molecules and the biology and virulence of this human pathogen.

## Figures and Tables

**Figure 1 jof-09-01102-f001:**
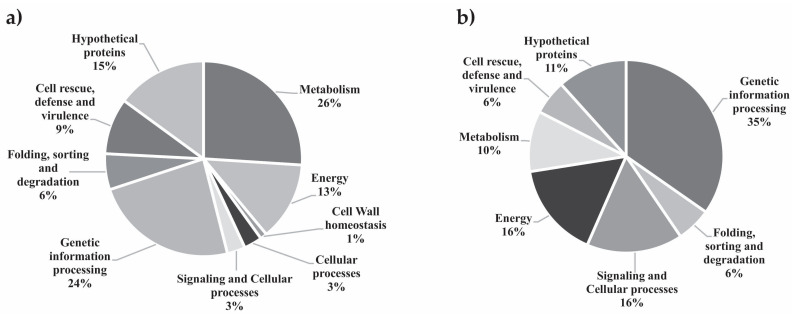
Functional classification of proteins regulated in *P. lutzii* identified by nanoUPLC-MS^E^. (**a**) Two hundred sixty-nine up-regulated proteins with their percentage (%) in each biological category. (**b**) Sixty-nine down-regulated proteins with their percentage (%) in each biological category. Biological processes involving the differentially expressed proteins were obtained using the Uniprot databases (http://www.uniprot.org/, accessed on 27 September 2022).

**Figure 2 jof-09-01102-f002:**
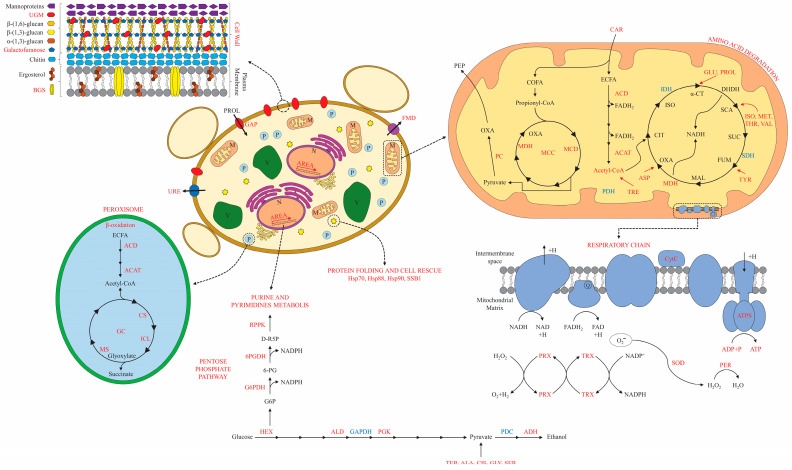
General overview of *P. lutzii* metabolism under NCR conditions. The metabolic pathways were determined based on the molecules identified by proteomic analysis in NCR-proline conditions. In red and blue, the molecules that were up-regulated and down-regulated, respectively, in NCR conditions are shown. Abbreviations: 6-PGDH: 6-phosphogluconate dehydrogenase; ACAT: acetyl-CoA acyltransferase; ACD: acyl-CoA dehydrogenase; ADH: alcohol dehydrogenase; ALA: alanine; ALD: aldolase; AREA: transcription factor AreA; ASP: asparagine; CAR: carnitine O-acetyltransferase; CIS: cysteine; COFA: odd-chain fatty acids; CS: citrate synthase; CytC: cytochrome C; DHDH: dihydrolipoyl dehydrogenase; ECFA: even-chain fatty acids; FMD: formamidase; G6PDH: glucose-6-phosphate 1-dehydrogenase; GAP: general amino acid permease; GAPDH: glyceraldehyde-3-phosphate dehydrogenase; GC: glyoxylate cycle; GLI: Glycine; GLU: glutamate; HSP70: heat shock protein 70; HSP88: heat shock protein Hsp88; HSP90: heat shock protein Hsp90; HX: hexokinase; ICL: isocitrate lyase; IDH: isocitrate dehydrogenase; ISO: isoleucine; MCC: methylcitrate cycle; MCD: 2-methylcitrate dehydratase; MDH: malate dehydrogenase; MET: methionine; MS: malate synthase; OXA: oxaloacetate; PC: pyruvate carboxylase; PDC: pyruvate decarboxylase; PDH: pyruvate dehydrogenase; PEP: phosphoenolpyruvate; PER: peroxidase; PGK: phosphoglycerate kinase; PRO: proline; PRX: mitochondrial peroxiredoxin; RPPK: ribose-phosphate pyrophosphokinase; SDH: succinate dehydrogenase; SER: serine; SOD: superoxide dismutase; SSB1: heat shock protein SSB1; THR: threonine; TRX: thioredoxin. TYR: tyrosine UGM: UDP-galactopyranose mutase; URE: urease; VAL: valine.

**Figure 3 jof-09-01102-f003:**
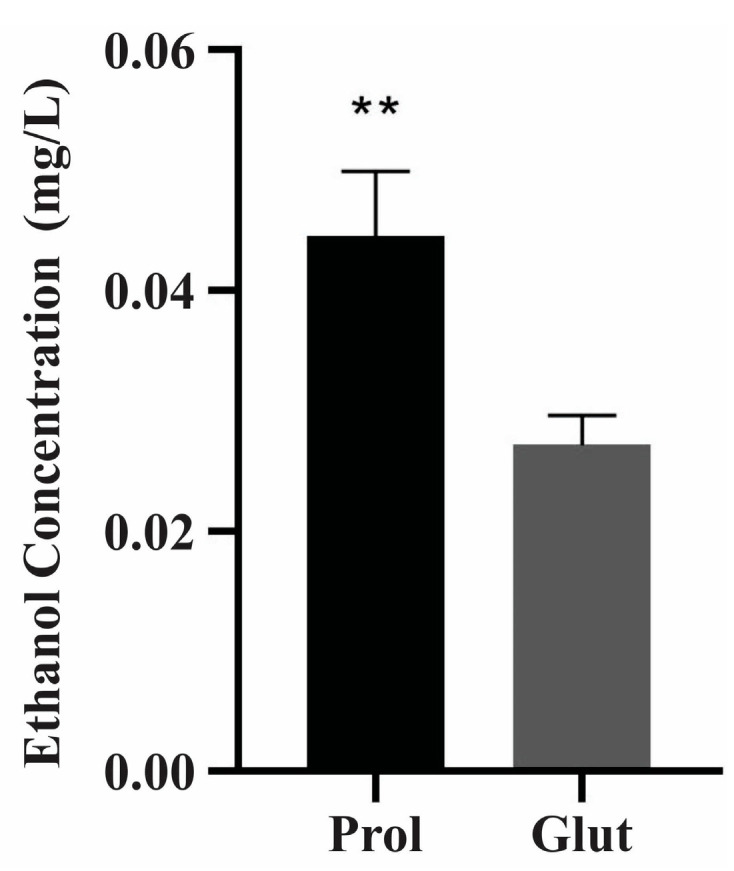
Ethanol measurement in NCR conditions. A total 2 × 10^8^ cells/mL were analyzed at 48 h under NCR conditions. Ethanol concentration was determined using an enzymatic detection kit (UV test for ethanol). The data are expressed as the mean ± standard deviation of biological triplicates analyzed in independent experiments and using the Student’s *t*-test, with *p* ≤ 0.01 (**) considered as statistically significant. Prol: proline; Glut: glutamine.

**Figure 4 jof-09-01102-f004:**
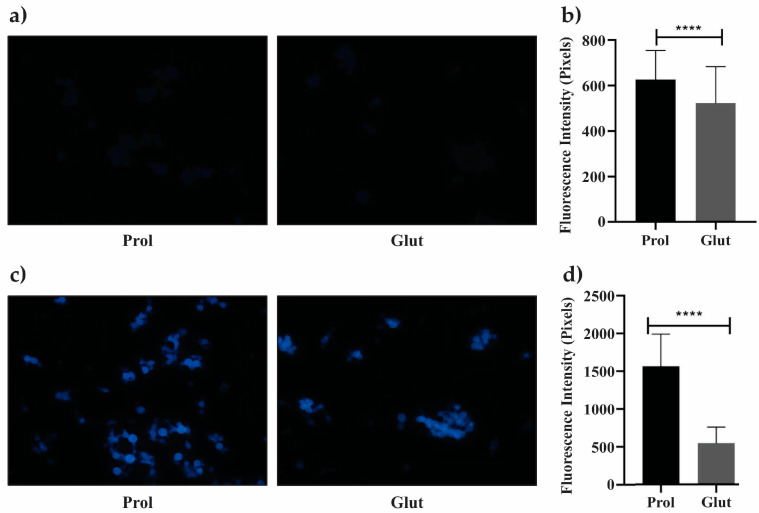
Cell wall remodeling in *P. lutzii* under proline conditions. Yeast cells were collected after 48 h of culture under NCR conditions, stained with aniline blue (AB) and calcofluor white (CFW) and visualized using an Axiocam MRc-Scope A1 fluorescence microscope. (**a**) Cells stained with AB in NCR-glutamine and NCR-proline conditions. (**b**) Fluorescence intensity of AB (in pixels) for yeast cells under NCR conditions. (**c**) Cells stained with CFW under NCR-glutamine and NCR-proline conditions. (**d**) Fluorescence intensity of CFW (in pixels) for yeast cells under NCR conditions was determined using AxioVision Software, determining the standard error in each analysis. Statistical comparisons were performed using Student’s *t*-test, with *p* ≤ 0.0001 (****), considered statistically significant. Prol: proline; Glut: glutamine; Magnification 40×.

**Figure 5 jof-09-01102-f005:**
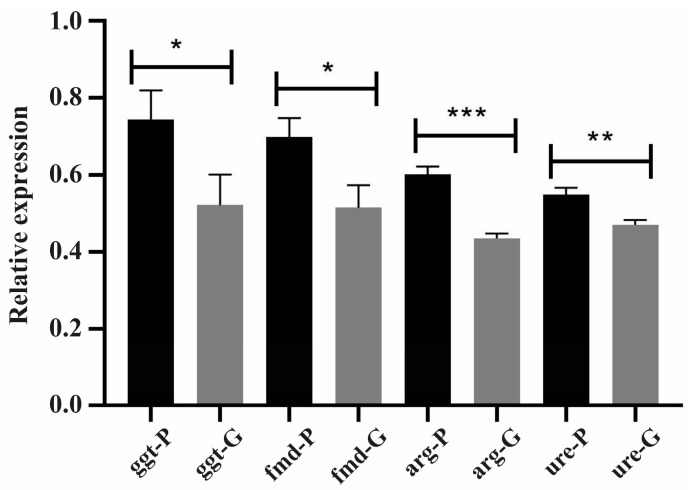
Transcriptional level of NCR-related genes under NCR conditions. The expression of NCR-related genes in *P. lutzii* yeast cells under NCR conditions was assessed through RT-qPCR, using relative quantification. The tubulin gene (PAAG_03031) served as the endogenous control, and the transcriptional levels of the genes coding for γ-glutamyl transpeptidase (*ggt*), formamidase (*fmd*), arginase (*arg*) and urease (*ure*) were analyzed. Data are expressed as the mean ± standard deviation of triplicates of independent experiments. Statistical analysis was performed through the Student’s *t*-test, demonstrating values of *p* ≤ 0.05 (*), *p* ≤ 0.01 (**) and *p* ≤ 0.001 (***), considered as statistically significant. P: proline; G: glutamine.

**Figure 6 jof-09-01102-f006:**
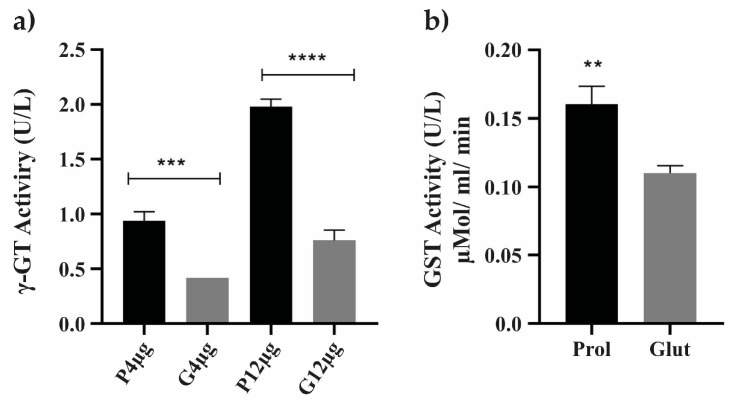
Enzymatic activity of enzymes related to cell rescue and virulence under NCR conditions. The assays were performed using soluble proteins obtained under NCR conditions. (**a**) The enzymatic activity of gamma-glutamyltransferase (γ-GT or GGT) was assessed using 4 μg and 12 μg of soluble proteins from NCR-proline and NCR-glutamine conditions. GGT specific activity (U/L) was calculated by measuring the number of nmol of ρNA released per liter of sample in the incubation time (min). (**b**) GST activity was measured using a total of 10 μg of soluble proteins obtained in NCR-proline and NCR-glutamine conditions. Student’s *t*-test was applied for statistical analysis, showing values of *p* ≤ 0.01 (**), *p* ≤ 0.001 (***) and *p* ≤ 0.0001 (****), considered as statistically significant. Prol: proline; Glut: glutamine.

**Figure 7 jof-09-01102-f007:**
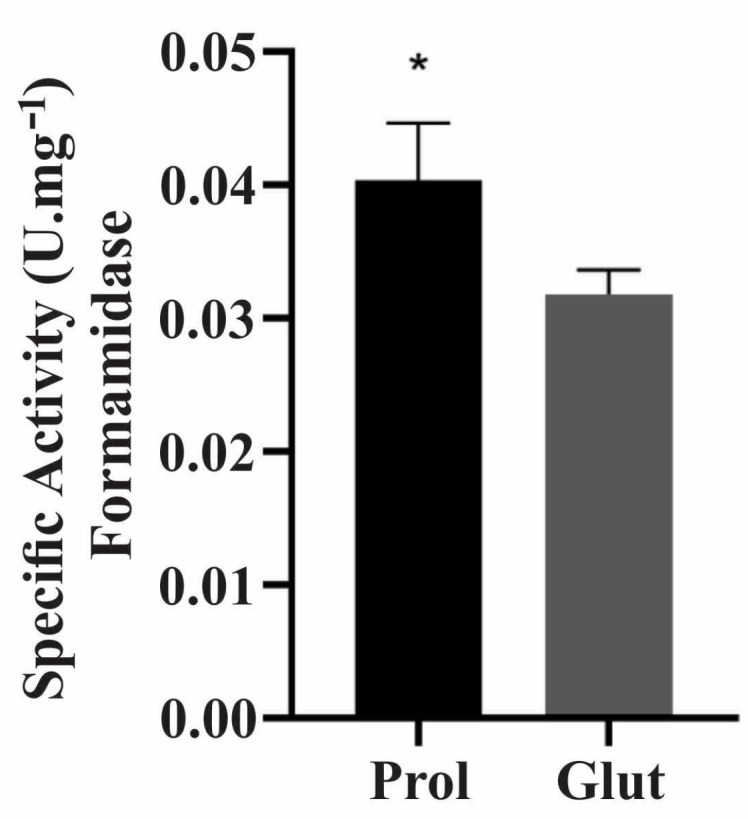
Enzymatic activity of formamidase in NCR conditions. The assays were performed with soluble proteins obtained in NCR conditions. Formamidase enzymatic activity in 10 μg of soluble proteins under NCR-proline and NCR-glutamine conditions; Student’s *t*-test was applied for statistical analysis showing values of *p* ≤ 0.05 (*), considered as statistically significant. Glut: glutamine; Prol: proline.

**Figure 8 jof-09-01102-f008:**
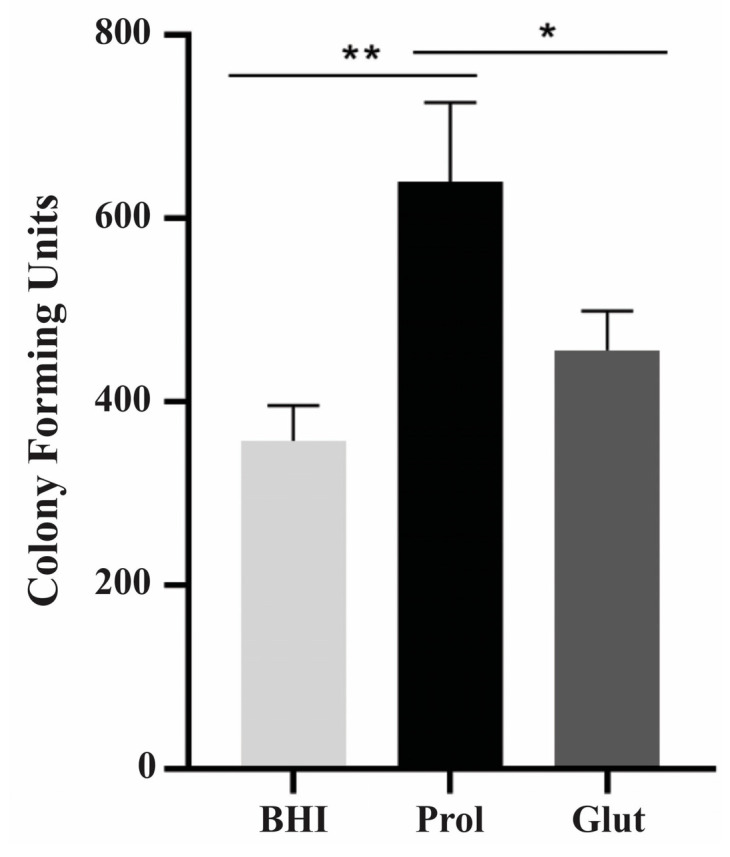
Survival of *P. lutzii* cells within macrophages. Prior to the ex vivo assays, *P. lutzii* yeast cells were cultured under NCR conditions for 48 h, followed by incubation with macrophages at a ratio of 1:5 (macrophages to yeast cells). The cells were then incubated for 12 and 24 h at 36 °C and in 5% CO_2_. The viable cell count was determined via quantification of colony-forming units (CFUs). The mean value and standard deviation were calculated from triplicates, and statistical analysis was conducted using Student’s *t*-test, showing values of *p* ≤ 0.05 (*) and *p* ≤ 0.01 (**), considered as statistically significant. BHI: brain heart infusion; Prol: proline; Glut: glutamine.

## Data Availability

Data are contained within the article.

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
