# Peer review of "Proteomics of Paracoccidioides lutzii: Overview of Changes Triggered by Nitrogen Catabolite Repression"

_jof, 2023, doi:10.3390/jof9111102_

Round 1

Reviewer 1 Report

Comments and Suggestions for Authors

In the manuscript ‘Proteomics of Paracoccidioides lutzii: Overview of changes triggered by Nitrogen Catabolite Repression’, the authors analyse proteome of P. lutzii during nitrogen catabolite repression (NRC)  and survival of the pathogen in human macrophages. Proteomic data of N-starved P. lutzii revealed that the yeast cells up-regulates several enzymes involved in beta-oxidation, gluconeogenesis and cell wall remodeling, allowing to survive in harsh conditions in macrophages.

Presented work is interesting to the other fungal researchers, working on Paracoccidioides sp.. I see few minor points concerning this work, as listed below:

In my opinion, the title should reflects the metabolism changes upon NCR response and yeast viability in macrophages after N-starvation, involving glyoxylate shunt.

section 2.4 – the main NCR marker, the GAP1 gene, should be used, as well

section 2.8 – the title would be more precise: Estimation of cell wall components in P. lutzii. CFW is a known agent, which binds to beta structures, such as polysaccharides (chitin, 1-4 glucan, cellulose), as well as beta sheet proteins. Hence, the applied method estimates sum of all beta polymers and the fluorescence signal may be overestimated.

Section 2.9 – the yeast cells should be prior opsonized. it is known that Paracoccidioides spp. (at least P. brasiliensis) are poorly phagocytosed by ‘stimulated’ mouse macrophages in culture (PMID: 421371). Hence, the statistical results may be doubtful in this case.

Row 436: Enzymatic...this sentence may be moved to M&M section.

Minor points:

Row 93: maydis

Comments on the Quality of English Language

The English language is fine. I don't see any errors

Reviewer 2 Report

Comments and Suggestions for Authors

In this manuscript, the authors used quantitative proteomics to investigate the NCR response of Paracoccidioides lutzii after growth on proline or glutamine as a nitrogen source. The experiments were designed and conducted in a systematic and convincing manner. The results were confirmed by other techniques, such as ethanol measurement, enzymatic activity assays, RT-qPCR, or western blot. The explanation of pathways was clear, and the conclusions are supported by the data. 

Minor points: 

# In line 171, as this is a proteomics paper, the description of how the authors performed proteomics should be included, even if there is a citation.

# Fig S1 seems to have only two lines.

# In line 261, why is the data not shown?

# In Fig 5, have the authors compared quantitative proteomics results with RT-qPCR results?

# Where does the author cite Fig 6 in the manuscript?

Author Response

Please, find the attached file with responses.
